# Multilayer Perceptron Network Optimization for Chaotic Time Series Modeling

**DOI:** 10.3390/e25070973

**Published:** 2023-06-24

**Authors:** Mu Qiao, Yanchun Liang, Adriano Tavares, Xiaohu Shi

**Affiliations:** 1School of Mathematics, Jilin University, Changchun 130021, China; 2Department of Industrial Electronics, School of Engineering, University of Minho, 4800-058 Guimares, Portugal; 3Key Laboratory of Symbol Computation and Knowledge Engineering of the Ministry of Education, College of Computer Science and Technology, Jilin University, 2699 Qianjin Street, Changchun 130012, China; ycliang@jlu.edu.cn; 4School of Computer Science, Zhuhai College of Science and Technology, Zhuhai 519041, China

**Keywords:** chaotic time series, multilayer perceptron network, generalized degrees of freedom, Akaike information criterion, maximal Lyapunov exponent

## Abstract

Chaotic time series are widely present in practice, but due to their characteristics—such as internal randomness, nonlinearity, and long-term unpredictability—it is difficult to achieve high-precision intermediate or long-term predictions. Multi-layer perceptron (MLP) networks are an effective tool for chaotic time series modeling. Focusing on chaotic time series modeling, this paper presents a generalized degree of freedom approximation method of MLP. We then obtain its Akachi information criterion, which is designed as the loss function for training, hence developing an overall framework for chaotic time series analysis, including phase space reconstruction, model training, and model selection. To verify the effectiveness of the proposed method, it is applied to two artificial chaotic time series and two real-world chaotic time series. The numerical results show that the proposed optimized method is effective to obtain the best model from a group of candidates. Moreover, the optimized models perform very well in multi-step prediction tasks.

## 1. Introduction

Some nonlinear dynamical systems present a seemingly irregular and random phenomenon, but they are actually produced by deterministic systems. For example, the famous “butterfly effect” concept describes how the flapping of a butterfly’s wings in Brazil could ultimately lead to a tornado in Texas that would not have happened otherwise. In other words, very tiny changes in conditions can result in very large, unpredictable responses, but this follows basic aerodynamic laws. This characteristic of nonlinear dynamical systems is called chaos. Chaotic dynamical systems are ubiquitous in nature, so they have been extensively studied to solve practical problems in different fields, such as financial systems analysis [1,2], power system behavior [3,4], information security [5,6], and the control of nonlinear systems [7,8]. In general, chaotic dynamical systems do not have an explicit dynamical equation and can only be understood through the available time series. Therefore, chaotic time series modeling is of great significance, but it is very difficult due to its complex, chaotic, nonlinear dynamics, such as internal randomness, nonlinearity, and long-term unpredictability.

Various researchers have shown great interest in chaotic time series analysis, finding many results, which could be classified into symbolic regression, polynomial model, model simulation or decomposition, and neural networks. For symbolic regression, Brandejsky used a GPA-ES system to study symbolic regression of deterministic chaotic systems [9], and Senkerik et al. proposed a novel tool for symbolic regression and analytical programming for the evolutionary synthesis of discrete chaotic systems [10]. The polynomial model is a widely used method. Lainscsek et al. used a truncated polynomial expansion involving successive derivatives of the measured time series according to an Ansatz library [11]. This procedure was introduced and improved by [12]. Han et al. utilized genetic programming (GP) and multi-objective optimization to identify chaotic systems using the nonlinear auto-regressive moving average with exogenous inputs (NARMAX) model representing the basis of the hierarchical tree encoding in GP [13]. Model simulation or decomposition is another type of method. Karimov et al. constructed a chaotic circuit from data to identify chaos systems [14]. Yang et al. introduced the Hankel Alternative View of Koopman analysis to decompose chaotic dynamics into a linear model with intermittent forcing [15]. The most widely used methods of chaotic time series analysis are neural network-related methods, which are classified into artificial neural networks (ANN) [16,17,18,19,20], fuzzy neural networks (FNN) [21,22,23,24], optimization algorithms with ANN [25,26,27,28], and wavelet neural networks (WNN) [29,30,31,32]. One can refer to [33] for a comprehensive review. Although there have been many attempts at chaotic time series analysis by neural network methods, including deep learning [16,34,35,36], there are still many issues to be resolved, such as low prediction accuracy and difficulty in determining the network topologies. Therefore, some scholars have studied the network optimization of chaotic time series analysis models. Xie et al. proposed an enhanced grey wolf optimizer (GWO) by designing four distinctive search mechanisms and then developed the evolving convolutional neural network–long short-term memory (CNN-LSTM) networks for time series analysis [37]. Huang et al. proposed an improved differential evolution (IDE) algorithm to optimize the topology of DHNNs consisting of CNN and GRU, including CNNs’ filter size and the number of hidden neurons of the GRU [38]. Focusing on the prediction of short-term traffic flow, Qian et al. found that the RBF neural network is better than the wavelet neural network, and they used the genetic algorithm to optimize the initial parameters [39]. Chen et al. presented a nonlinear ensemble of partially connected neural networks for short-term load forecasting, in which the genetic algorithm is used to generate diverse and effective neural networks, and a novel pruning method was developed to optimize the partially connected neural networks [40]. Kao et al. studied a Takagi-Sugeno-Kang (TSK)-type self-organizing fuzzy neural network, which not only generates and prunes the learning algorithm automatically but also adjusts the parameters of existing fuzzy rules [41]. Though all the above methods work on network structure optimization, they are empirical and have no strict theoretical basis.

The Akaike information criterion (AIC) [42] is a popular tool to evaluate a linear auto-regression model. The idea of AIC has been extended to nonlinear models, especially the neural networks based on the generalized degrees of freedom [43,44,45]. However, when employing the Akaike information criterion based on the generalized degrees of freedom (AIC_g_) to evaluate the MLP network, the generalized degrees of freedom must be estimated. For a linear regression model, the degrees of freedom are the number of variables in the model, which is the sum of the sensitivities of each fitted value with respect to the corresponding observed values. For nonlinear models, Ye [45] defined a new quantity known as the generalized degrees of freedom (GDF) and proved that the GDF provides an unbiased estimation of the error variance. However, it is very difficult to calculate the GDF using the original definition. A Monte Carlo method is therefore suggested to calculate the GDF for a nonlinear model. This approach requires estimating the nonlinear model many times and is therefore time-consuming.

Recently, a new method for the estimation of the generalized degrees of freedom for the RBF-type network has been introduced [44]. Following that method, the generalized hat matrix for the nonlinear model is defined in this study; its trace is an estimation of the generalized degrees of freedom. The proposed estimation method simulates the model procedure once only and therefore is faster than the Monte Carlo method. Using the AIC_g_, the performances of the MLP networks for different network topologies are evaluated, and the optimal MLP network for the chaotic time series is then selected. The model-selecting method is applied to four chaotic time series, two artificial time series and two real-world time series. By estimating the maximal Lyapunov exponents and forecasting the time series, the numerical results show that the MLP network selected by the proposed method behaves better in terms of the analytic and prediction capabilities than other models. The main contributions of the paper are the following three points:Present an approximation method to calculate the generalized degree of freedom of MLP, and then obtain its AIC_g_.Develop a multilayer perceptron network optimization method for chaotic time series analysis by designing the AIC_g_ as the loss function of MLP.Propose an overall framework for chaotic time series analysis, including phase space reconstruction, model training, and model selection modules.

This paper is arranged as follows: In Section 2, some backgrounds are introduced, including phase space reconstruction of chaotic systems, the Akaike information criterion of nonlinear systems, and generalized degrees of freedom of MLP. In Section 3, our proposed modeling selection framework for MLP networks is described in detail. In Section 4, the proposed method is applied to four chaotic time series, and the results are presented and analyzed. The last section gives a brief conclusion.

## 2. Backgrounds

### 2.1. Phase Space Reconstruction of Chaotic Systems

Suppose that the chaotic system is an n-dimensional system given by
(1)$$\left\{\begin{array}{c} x_{1}^{\left(i+1\right)}=f_{1}\left(x_{1}^{\left(i\right)},x_{2}^{\left(i\right)},\cdots ,x_{n}^{\left(i\right)}\right)\\ x_{2}^{\left(i+1\right)}=f_{2}\left(x_{1}^{\left(i\right)},x_{2}^{\left(i\right)},\cdots ,x_{n}^{\left(i\right)}\right)\\ \cdots \\ x_{n}^{\left(i+1\right)}=f_{n}\left(x_{1}^{\left(i\right)},x_{2}^{\left(i\right)},\cdots ,x_{n}^{\left(i\right)}\right) \end{array}\right.$$
where *X* = (*x*_1_, *x*_2_, …, *x_n_*) is an n-dimension vector.

In general, the real observed chaotic time series is only one or a few dimensional state sequences. In order to analyze the chaotic time series, it is necessary to reconstruct the phase space of the chaotic system from the observed low-dimensional time series and restore the chaotic attractor of the original phase space in this high-dimensional embedded space. Takens’ theorem is the theoretical basis for phase space reconstruction [46].
**Theorem 1.** *(Takens’ theorem* [46]*): Suppose M is an m-dimensional compact manifold, the map φ: M→M is a smooth differential isomorphism, y: M→R is a smooth function, and the map Φ _(φ, y)_: M→R^2m+1^ is defined by*
*Φ*_(*φ*, *y*)_ = (*y*(*x*), *y*(*φ*(*x*)), *y*(*φ^2^*(*x*)), …, *y*(*φ^2m^* (*x*)))(2)
Then, *Φ*
_(*φ*, *y*)_ is an embedding from *M* to *R*^2*m*+1^.

This theorem states that for an *m*-dimensional chaotic dynamic system, as long as the embedding dimension is greater than or equal to 2*m* + 1, the chaotic attractor can be restored in the embedding space; that is, a system with the same dynamic properties as the original chaotic system can be obtained in the embedding space. This mathematically guarantees that a phase space equivalent to the original system in the topological sense can be reconstructed from a lower-dimensional or even one-dimensional chaotic time series. The coordinate delay method is commonly used to reconstruct the phase space of the chaotic system. The essence is to construct *m*-dimensional phase space vectors from an observed one-dimensional time series {*x_i_*} (*i* ∈ [1, *N*]) according to a series with different time delays, namely, [47]
(3)$$X=\left[\begin{array}{c} X_{1}\\ X_{2}\\ \vdots \\ X_{N-\left(m-1\right)\tau } \end{array}\right]=\left[\begin{array}{cccc} x_{1} & x_{1+\tau } & \cdots & x_{1+\left(m-1\right)\tau }\\ x_{2} & x_{2+\tau } & \cdots & x_{2+\left(m-1\right)\tau }\\ \vdots & \vdots & \ddots & \vdots \\ x_{N-\left(m-1\right)\tau } & x_{N-\left(m-2\right)\tau } & \cdots & x_{N} \end{array}\right]$$
where *N* is the time series length, *m* the embedding dimension, and *τ* the time delay. According to Takens’ theorem, if the choice of the embedding dimension *m* and time delay *τ* is appropriate, the “orbit” of the reconstructed phase space in the embedded space is equivalent to the dynamics of the original system (Equation (1)) in the sense of differential isomorphism. This method is called the delay coordinate embedding approach. In this approach, the determination of the values of *m* and *τ* is the key to phase space reconstruction.

### 2.2. Akaike Information Criterion of a Nonlinear System

Assume that a nonlinear system is modeled by the function [44]
*y* = *f* (**X**, **α**) + *ε*,(4)
where **X** = (*x*_1_, *x*_2_,…,*x_d_*) ∈ **R**^d^ is the input, **α** = (*α*_1_, *α*_2_,…, *α_K_*) ∈ **R**^K^ is the model parameter, *ε*∈ **R** is the random noise, and *y* ∈ **R** is the output. Assume also that we have *N* input samples **X** = (**X**_1_, **X**_2_,…, **X***_N_*) and *N* output observers **Y** = ($\hat{y}$_1_, $\hat{y}$_2_,…, $\hat{y}$*_N_*), and denote the function output of **X**_n_ as *y*_n_ = *f* (**X**_n_, **α**) + *ε*. Then, the residual sum of squares (RSS) of the MLP network is [44]
$$RSS=\sum_{j=1}^{N}{\left(y_{n}-{\hat{y}}_{n}\right)}^{2}$$
For a nonlinear model, the quadratic loss of the model can be estimated by [30]
(5)$$A=RSS-N\sigma^{2}+2n_{g}\sigma^{2}$$
where *σ* is the variance of the observation error and *n_g_* is the generalized degrees of freedom. The mean of *RSS*/(*N* − *n_g_*) was proven to be equal to the variance *σ*^2^; hence, the quadratic loss can be approximated by [45]
$$\begin{array}{cc} A & \approx RSS-N\sigma^{2}+2n_{g}\frac{RSS}{N-n_{g}}\;\\ & =RSS\left(\frac{N+n_{g}}{N-n_{g}}\right)-N\sigma^{2}\\ & =RSS{\left(\frac{N}{N-n_{g}}\right)}^{2}(1-\left(\frac{n_{g}}{N}{)}^{2}\right)-N\sigma^{2} \end{array}$$
Since *n_g_* is much smaller than *N* and *σ* is a constant, another criterion *G* could be obtained by [48]
(6)$$G=RSS{\left(\frac{N}{N-n_{g}}\right)}^{2}$$
Taking the logarithm for Equation (6), we can obtain
$$\ln(G)=ln(RSS)-2\ln(1-\frac{n_{g}}{N})\;=\ln(RSS/N)+\ln(N)+2n_{g}/N+{\left(\frac{n_{g}}{N}\right)}^{2}+\ldots \;\approx \ln(RSS/N)+2n_{g}/N+\ln(N)$$
It should be noted that the above derivation makes use of the Taylor expansion. Then, we arrive at the form of the Akaike information criterion for an MLP network,
(7)$${\mathrm{AIC}}_{g}=ln(RSS/N)+2n_{g}/N$$
which is based on the concept of entropy and able to weigh the complexity of the estimated model and the goodness of the fitted data of the model.

### 2.3. Generalized Degrees of Freedom (GDF)

In order to calculate the Akaike information criterion of an MLP network according to Equation (7), the generalized degrees of freedom (GDF) *n_g_* should be obtained in advance. Xu et al. developed a fast estimation method for GDF of a nonlinear system [44], which is briefly described in this section for the convenience of the reader.

GDF is defined as “the sum of sensitivity of each fitted value to the perturbation in the corresponding observed value”, which can measure the complexity of a nonlinear model [45]. Xu et al. proposed an approximated method to obtain the GDF of a nonlinear system [44].

**Theorem 2.** 
*If a nonlinear model f (**X**, **α**) is adequate in fitting the data and has bounded second order derivatives, the generalized degrees of freedom for the nonlinear system can be approximated by*


(8)$$n_{g}\approx trace\left(H\right)$$
where ***H*** is the generalized hat matrix of the nonlinear system, defined by
(9)$$H=M^{T}{\left(M^{T}M\right)}^{-1}M$$
where ***M*** is the gradient matrix of parameters.
(10)$$M=\left[\begin{array}{ccccc} \frac{\partial f\left(X_{1},\alpha \right)}{\partial \alpha_{1}} & \frac{\partial f\left(X_{1},\alpha \right)}{\partial \alpha_{2}} & \cdots & \frac{\partial f\left(X_{1},\alpha \right)}{\partial \alpha_{k-1}} & \frac{\partial f\left(X_{1},\alpha \right)}{\partial \alpha_{k}}\\ \frac{\partial f\left(X_{2},\alpha \right)}{\partial \alpha_{1}} & \frac{\partial f\left(X_{2},\alpha \right)}{\partial \alpha_{2}} & \cdots & \frac{\partial f\left(X_{2},\alpha \right)}{\partial \alpha_{k-1}} & \frac{\partial f\left(X_{2},\alpha \right)}{\partial \alpha_{k}}\\ \vdots & \vdots & \cdots & \vdots & \vdots \\ \frac{\partial f\left(X_{N-1},\alpha \right)}{\partial \alpha_{1}} & \frac{\partial f\left(X_{N-1},\alpha \right)}{\partial \alpha_{2}} & \cdots & \frac{\partial f\left(X_{N-1},\alpha \right)}{\partial \alpha_{k-1}} & \frac{\partial f\left(X_{N-1},\alpha \right)}{\partial \alpha_{K}}\\ \frac{\partial f\left(X_{N},\alpha \right)}{\partial \alpha_{1}} & \frac{\partial f\left(X_{N},\alpha \right)}{\partial \alpha_{2}} & \cdots & \frac{\partial f\left(X_{N},\alpha \right)}{\partial \alpha_{k-1}} & \frac{\partial f\left(X_{N},\alpha \right)}{\partial \alpha_{K}} \end{array}\right]$$

Note that the MLP basis functions clearly have bounded second order derivatives. Hence, Theorem 1 can be applied to calculate the generalized degrees of freedom for an MLP model.

## 3. Modeling Selection for MLP Networks

### 3.1. Pipeline of the Modeling Selection

MLP networks are effective tools for chaotic time series predictions. However, the network topology seriously affects the performance of algorithms. Therefore, a pipeline of the modeling selection for MLP networks is proposed in this paper, as shown in Figure 1. For a chaotic time series dataset, the phase space reconstruction should be performed in advance. Next, a candidate MLP structure pool is generated. Then, each structure is trained to determine its parameters. It should be noticed that the Akaike information criterion is used as the loss function for training. When the training process is completed, the MLP network structure with the least Akaike information criterion value is selected as the best one, that is, the final prediction model. The phase space reconstruction process is described in Section 3.2, and the Akaike information criterion computation is detailed in Section 3.3, respectively.

### 3.2. Phase Space Reconstruction

According to Takens’ theorem, the *m*-dimensional phase space of the chaotic system could be reconstructed from an observed one-dimensional time series by the delay coordinate embedding approach (Equation (3)). The key to the method is how to determine the embedding dimension *m* and time delay *τ*. To address this issue, many algorithms have been proposed: For example, the mutual information method [48] and the autocorrelation coefficients method [49] are two popular approaches to determine the time delay *τ*; the false nearest neighbor method [50] and its improved version the Cao method [51] are often used to determine the embedding dimension *m*; and Uzal et al. proposed a noise amplification approach able to handle these two tasks simultaneously [52]. In our pipeline, the time delay *τ* is determined by the mutual information method [48], and the embedding dimension *m* is selected by the Cao method [51].

We denote the observed one-dimensional time series {*x_i_*} as a discrete system *S*. The amount of information contained in this system can be characterized in terms of entropy:(11)$$H\left(S\right)=\sum_{s\in S}p_{s}logp_{s}$$
where *p_s_* represents the probability of *s* occurring. We denote the delayed time series {*x_i_*_+*τ*_} as another discrete system *Q*. Then, the joint entropy of the systems *S* and *Q* could be defined by.
(12)$$H\left(S,Q\right)=\sum_{s\in S,q\in Q}p_{sq}logp_{sq}$$
where *p_sq_* represents the joint probability of *s* and *q* appearing in systems *S* and *Q*. Thereby, the mutual information between the two systems could be defined by
(13)$$I\left(S,Q\right)=H\left(S\right)+H\left(Q\right)-H\left(S,Q\right)$$
which can be understood as the amount of overlapping information between these two systems. After fixing the original system *S*, system *Q* is determined by the time delay *τ*. Taking the time delay *τ* as a variable, the mutual information *I*(*S*, *Q*) can be treated as a function of *τ*, denoted as *I*(*τ*). In order to make the correlation between the delayed time series and the original series as small as possible, the minimum point of *I*(*τ*) is taken as the final time delay:(14)$$\tau^{*}=\arg\min_{\tau }I\left(\tau \right)$$

Considering that we want a small time delay, the first local minimum point is usually taken as the final result in the real applications. The pseudocode of the time delay determination is given by Algorithm 1.

**Algorithm 1**: Time delay determination
Input: one-dimensional time series *S* = {*x_i_*}; time delay max value τ_max_; iteration number *N*
Output: best time delay τ^*^1$\mathrm{compute}\;\mathrm{the}\;\mathrm{entropy}\;\mathrm{of}\;S:\;H\left(S\right)=\sum_{s\in S}p_{s}log\;p_{s}$2for i = 1: *N*
3    time delay τ = *i* × τ_max_/*N*4    time delayed system *Q* = {*x_i+__τ_*}5    $\mathrm{compute}\;\mathrm{the}\;\mathrm{entropy}\;\mathrm{of}\;Q:\;H\left(Q\right)=\sum_{q\in Q}p_{q}log\;p_{q}$
6  $\;\mathrm{compute}\;\mathrm{the}\;\mathrm{joint}\;\mathrm{entropy}\;\mathrm{of}\;S\mathrm{and}Q:\;H\left(S,Q\right)=\sum_{s\in S,q\in Q}p_{sq}log\;p_{sq}$
7  $\;\mathrm{compute}\;\mathrm{the}\;\mathrm{mutual}\;\mathrm{information}\;\mathrm{of}\;S\mathrm{and}Q:\;I(\tau )=H\left(S\right)+H\left(Q\right)-H\left(S,Q\right)$
8end for9τ^*^
$=arg\;\underset{\tau }{\min\;}I\left(\tau \right)$


A chaotic time series is a projection in one- or lower-dimensional space of a deterministic dynamical system’s trajectory in a high-dimensional phase space. The projection modifies the topological properties of the motion trajectory, resulting in “disorder” characteristic of chaotic systems. Thus, we can observe the change in the “nearest neighbor” distances when the embedding dimension increases, and it stops when it no longer changes.

Let *x_i_* be a point of the original time series, and *X_i_*(*m*) = (*x_i_*, *x_i_*_+*τ*_,…, *x_i_*_+(*m*−1)*τ*_) and *X_i_*_(*m*+1)_ = (*x_i_*, *x_i_*_+*τ*_,…, *x_i_*_+*mτ*_) be its mapping points in the *m*-dimensional and (*m* + 1)-dimensional embedding spaces. Denote *X_n_*_(*i,m*)_ (*m*) as the nearest neighbor to *X_i_*(*m*) in the embedding space, i.e.,
(15)$$n\left(i,m\right)=\underset{j}{arg}\min∥X_{i}^{\left(m\right)}-X_{j}^{\left(m\right)}∥{}_{2}$$
Then, the change of the distance of this “nearest neighbor” pair when the embedding dimension increases from *m* to *m* + 1 could be represented by
(16)$$a\left(i,m\right)=\frac{{∥X_{i}^{\left(m+1\right)}-X_{n\left(i,m\right)}^{\left(m+1\right)}∥}_{\infty }}{{∥X_{i}^{\left(m\right)}-X_{n\left(i,m\right)}^{\left(m\right)}∥}_{\infty }}\;i=1,2,\cdots ,n-m\tau$$
Thus, the average distance change of the “nearest neighbor” pairs could be expressed as
(17)$$E\left(m\right)=\frac{\sum_{i=1}^{n-m\tau }a\left(i,m\right)}{n-m\tau }\;\;m=1,2,\cdots .$$
It can be understood as the average scale of the nearest neighbor distance of all mapped points in *m*-dimensional embedding space after mapping to (*m* + 1)-dimensional space. Thus, we could define
(18)$$E_{1}\left(m\right)=\frac{E\left(m+1\right)}{E\left(m\right)\;}\;\;\;m=1,2,\cdots .$$

If the time series is not random, there exists an embedding dimension *d*^*^ satisfying the Takens’ theorem, such that when *m* > *d*^*^, *E*_1_(*m*) no longer changes; otherwise, *E*_1_(*m*) continues to increase.

To further ensure the reliability of the method, the following judgment criterion has been added:(19)$$E^{*}\left(m\right)=\frac{\sum_{i=1}^{n-d\tau }\left|x\left(i+m\tau \right)-x\left(n\left(i,m\right)+m\tau \right)\right|}{n-m\tau }\;\;\;m=1,2,\cdots ,$$
(20)$$E_{2}\left(m\right)=\frac{E^{*}\left(m+1\right)}{E^{*}\left(m\right)\;\;}\;\;\;\;m=1,2,\cdots .$$
Since there is no correlation between random sequence data, *E*_2_(*m*) ≡ 1, and for deterministic sequences, there must exist *m*, such that *E*_2_(*m*) ≠ 1. The pseudocode of the embedding dimension determination is given by Algorithm 2.

**Algorithm 2**: Embedding dimension determination
Input: one-dimensional time series *S* = {*x_i_*,…, *x_n_*}, time delay *τ*;    max embedding dimension *M*, threshold *ε.*
Output: embedding dimension *m*^*^1for k=1: n−mτ2for *m* = 1: *M*
3   *E*(*m*) = 0; *E*^*^*(m*) *=* 0; *m*^*^
*=* 0;4   for *i* = 1: *n − mτ*5     
$n\left(i,m\right)=\underset{j}{arg}\min X_{i}^{\left(m\right)}-X_{j}^{\left(m\right)}{}_{2}$
6     
$a\left(i,m\right)=\frac{X_{i}^{\left(m+1\right)}-X_{n\left(i,m\right)}^{\left(m+1\right)}{}_{\infty }}{X_{i}^{\left(m\right)}-X_{n\left(i,m\right)}^{\left(m\right)}{}_{\infty }}$
7     *E*(*m*) = *E*(*m*) + *α*(*i*, *m*)/(*n − mτ*)8     *E*^*^ (*m*) = *E*^*^ (*m*) + *|x*(*i + mτ*)* − x*(*n*(*i,m*)* + mτ)|*/(*n − mτ*)9   end for10 end for11 *E*_1_(1) = *E*(2)*/E*(1)12 *E*_2_(1) = *E*^*^(2)*/E*^*^(1)13 for *m* = 2: *M* − 114   *E*_1_(*m*) = *E*(*m* + 1)*/E*(*m*)15   *E*_2_(*m*) = *E*^*^(*m* + 1)*/E*^*^(*m*)16   if (*|E*_1_(*m*) − *E*_1_(*m* − 1)*|* < *ε*) and (*|E*_2_(*m*) − 1*|* > *ε*)17     *m*^*^ = *m*18   end if19 end for

### 3.3. Calculation of MLP Networks’ Akaike Information Criterion (AICg)

The Akaike information criterion (AIC_g_) offers a relative performance measure when nonlinear models are used to represent the process that actually generates the data. Hence, the AIC_g_ provides a means for model selection for an MLP network. In practice, several candidate MLP networks with different neuron sizes are built and trained using the same sample set. The residual sum of squares for every candidate network is then obtained. In order to apply the AIC_g_, the GDF for every candidate network is needed. According to Equation (8), the GDF can be estimated easily by the gradient matrix of the parameters. In this study, using the idea of back-propagation, we introduce a method to calculate the gradient matrix of the parameters for an MLP network.

Consider a multilayer perceptron network with one input layer, *K* hidden layers, and one output layer. There are *M_k_* neurons in the kth hidden layer. In order to address the bias in the MLP network freely, we first expand the input vector and the neurons. Denote the expanding input vector by
(21)$$U^{\left(0\right)}={\left(x_{n,1},x_{n,2},\cdots ,x_{n,d},1\right)}^{T}$$
Then, the feed vector in the first hidden layer is
(22)$$p^{\left(1\right)}=w^{\left(1\right)}U^{\left(0\right)}$$
The parameter matrix **w**^(1)^ is given by
(23)$$w^{\left(1\right)}=\left(\begin{array}{ccccc} w_{11}^{\left(1\right)} & w_{12}^{\left(1\right)} & \cdots & w_{1d}^{\left(1\right)} & \beta_{1}^{\left(1\right)}\\ w_{21}^{\left(1\right)} & w_{22}^{\left(1\right)} & \cdots & w_{2d}^{\left(1\right)} & \beta_{2}^{\left(1\right)}\\ \vdots & \vdots & \cdots & \vdots & \vdots \\ w_{M_{1}1}^{\left(1\right)} & w_{M_{1}2}^{\left(1\right)} & \cdots & w_{M_{1}d}^{\left(1\right)} & \beta_{M_{1}}^{\left(1\right)} \end{array}\right)$$
where *d* is the dimension of the input vector, *M*_1_ is the number of the first hidden layer nodes, and *β_i_*^(1)^ is the bias of the *i*th node, respectively.

In the same way, the input vector in the *k*th hidden layer can be extended by
(24)$$\begin{array}{cc} U\left(k-1\right) & =(o_{1}^{\left(k-1\right)},o_{2}^{\left(k-1\right)},\cdots ,o_{M_{k-1}}^{\left(k-1\right)},1)T\\ & =(f(p_{1}^{\left(k-1\right)},f(p_{2}^{\left(k-1\right)},\cdots ,f(p_{M_{k-1}}^{\left(k-1\right)}),1)T \end{array}$$
The feed vector in the *k*th hidden layer is then given by
(25)$$p^{\left(k\right)}=w^{\left(k\right)}U^{\left(k-1\right)}$$
Finally, the input vector for the output layer is
(26)$$U^{\left(K\right)}={\left(o_{1}^{\left(K\right)},o_{2}^{\left(K\right)},\cdots ,o_{M_{K-1}}^{\left(K\right)},1\right)}^{T}$$
The feed value for the output layer is
(27)$$p^{\left(K+1\right)}=w^{\left(K+1\right)}U^{\left(K\right)}$$
The output of the MLP network is then given by
(28)$${\hat{y}}_{n}=g\left(p^{\left(K+1\right)}\right)$$
Let the activation function ***f*** in the hidden layer be the sigmoid function
(29)$$f\left(s\right)=\frac{1}{1+\exp(-s)}$$
Then
(30)$$f^{\prime }\left(s\right)=f\left(s\right)\left(1-f\left(s\right)\right)$$
Let the activation function ***g*** in the output layer be the linear function
(31)$$g\left(s\right)=s$$
Then
(32)$$g^{\prime }\left(s\right)=1$$

Denote the MLP network as a function
(33)$$y_{n}=F\left(X_{n},w^{\left(1\right)},w^{\left(2\right)},\cdots ,w^{\left(K+1\right)}\right)$$
Let $Q^{\left(k\right)}(i,j)$ be a matrix with the same dimension as $w^{\left(k\right)}$ where all its elements are zero except $q_{ij}=1$. Furthermore, define matrix $O^{\left(k\right)}$ by
(34)$$O^{\left(k\right)}=\left(\begin{array}{cccc} o_{1}^{\left(k\right)}\left(1-o_{1}^{\left(k\right)}\right) & 0 & \cdots & 0\\ 0 & o_{2}^{\left(k\right)}\left(1-o_{2}^{\left(k\right)}\right) & \cdots & 0\\ \vdots & \vdots & \cdots & \vdots \\ 0 & 0 & \cdots & o_{M_{k}}^{\left(k\right)}\left(1-o_{M_{k}}^{\left(k\right)}\right)\\ 0 & 0 & \cdots & 0 \end{array}\right)$$
Then, we have
(35)$$\frac{dU^{\left(k+1\right)}}{dy}=O^{\left(k\right)}\frac{d}{dy}\left(w^{\left(k\right)}U^{\left(k-1\right)}\right)$$
Hence, the derivative of the MLP function with respect to $w_{j}^{\left(K+1\right)}$ is given by
(36)$$\frac{\partial F}{\partial w_{j}^{\left(K+1\right)}}=\frac{\partial }{\partial w_{j}^{\left(K+1\right)}}\left(w^{\left(K+1\right)}U^{\left(K\right)}\right)=Q^{\left(K+1\right)}\left(1,j\right)U^{\left(K\right)},\;j=1,2,\cdots ,M_{K}+1$$
Furthermore, the derivative of the MLP function with respect to $w_{j,m}^{\left(K\right)}$ is given by
(37)$$\frac{\partial F}{\partial w_{j,m}^{\left(K\right)}}=w^{\left(K+1\right)}\frac{\partial }{\partial w_{j,m}^{\left(K\right)}}U^{\left(K\right)}=w^{\left(K+1\right)}O^{\left(K\right)}Q^{\left(K\right)}\left(j,m\right)U^{\left(K-1\right)}$$
By the same discussion, we obtain the derivative with respect to $w_{j,m}^{\left(k\right)}$,
(38)$$\frac{\partial F}{\partial w_{j,m}^{\left(k\right)}}=\left(\prod_{n=k}^{K}w^{\left(n+1\right)}O^{\left(n\right)}\right)Q^{\left(k\right)}\left(j,m\right)U^{\left(k-1\right)}$$
Thus, the gradient matrix of parameters ***M*** is obtained, and then the generalized hat matrix ***H*** can be calculated according to Equation (9). The trace of the generalized hat matrix is an estimator of the GDF *n_g_* for the MLP network according to Equation (8).

Using the GDF estimated by the generalized hat matrix, the AIC_g_ for each MLP candidate is calculated by
(39)$${\mathrm{AIC}}_{g}=\ln(RSS/N)+2n_{g}/N=\ln(\sum_{j=1}^{N}{\left(y_{n}-{\hat{y}}_{n}\right)}^{2}/N)+2n_{g}/N$$

The pseudocode of the calculation of MLP networks’ Akaike information criterion is given by Algorithm 3.

**Algorithm 3**: Calculation of MLP networks’ AIC_g_
Input: ANN model ANN = {w_i,m_^(k)^}, embedded time series data G = {X_1_,…,X_N_}, iteration number N_Max.
Output: MLP networks’ Akaike’s information criterion AIC_g_
1 for i = 1: N_Max2   for n = 1: N3      
${\hat{y}}_{n}=\mathrm{ANN}\left(X_{n}\right)=g\left(p^{\left(K+1\right)}\right)$
4      
$\frac{\partial F}{\partial w_{j,m}^{\left(k\right)}}=\left(\prod_{n=k}^{K}w^{\left(n+1\right)}O^{\left(n\right)}\right)Q^{\left(k\right)}\left(j,m\right)U^{\left(k-1\right)}$
5      
$H=M^{T}{\left(M^{T}M\right)}^{-1}M$
6      
$n_{g}\approx \mathrm{trace}\left(H\right)$
7      
${\mathrm{AIC}}_{g}=\ln(\sum_{j=1}^{N}{\left(y_{n}-{\hat{y}}_{n}\right)}^{2}/N)+2n_{g}/N$
8   end for9 end for

## 4. Experiments

### 4.1. Benchmark Datasets

In order to verify the effect of the proposed method, it is applied to simulate four chaotic time series: two artificial chaotic time series and two real-world time series. The two artificial benchmarks are the Hénon map [53] and the Lorenz map [54], and the two real ones are sunspots time series [55] and SST data [56], respectively.

Hénon map: The Hénon map [53] is given as
(40)$$y_{n+1}=1-ay_{n}+by_{n-1}$$

It has a chaotic attractor when *a* = 1.4 and *b* = 0.3. A time series of length 1500 is generated with initial conditions *y*_−1_ = 1.005 and *y*_0_ = −0.3032. The maximal Lyapunov exponent (MLE) of the Henon attractor is about 0.419.

Lorenz equation: The Lorenz map is [54] defined as
(41)$$\left\{\begin{array}{c} \frac{dx}{dt}=\sigma \left(y-x\right)\\ \frac{dy}{dt}=-xz+rx-y\\ \frac{dy}{dt}=xy-bz \end{array}\right.$$
It becomes chaotic for σ = 10, *r* = 28, and *b* = 8/3. Equation (41) is a three-dimensional differential system, which can be solved numerically using the fourth order Runge-Kutta method with a time step of 0.01 and initial conditions of
(42)$$x\left(0\right)=2.4151,y\left(0\right)=3.6936,z\left(0\right)=15.1426$$
A time series of length 1500 is generated with sampling frequency 0.2,
$$x_{n}=x\left(0.2n\right),\;1\leq n\leq 1500$$
The MLE for the Lorenz attractor is about 0.91. Since the sampling frequency is 0.2, the maximal Lyapunov exponent for the time series is therefore about 0.182.

Sunspots time series: Sunspots are temporary phenomena on the surface of the Sun (the photosphere) that are visible as dark spots compared to the surrounding regions. The sunspots index has been used since 1749. The monthly averaging sunspots index considered in this study is from 1749 to 2009 with a sample size of 3132. The dataset can be downloaded from the website: psl.noaa.gov/gcos_wgsp/Timeseries/SUNSPOT/ (accessed on 5 June 2023). The sunspots data are a nonlinear time series, which have been discussed as a chaotic time series by many authors [55].

SST data: The sign and magnitude of equatorial Pacific sea-surface temperature anomalies (SSTA) provide a measure of the ENSO phase and strength. The El Niño phase produces sea-surface temperatures in the equatorial eastern Pacific that are anomalously warm with respect to the mean seasonal cycle, while La Niña conditions produce anomalously cold sea-surface temperature conditions. The S index of Wright [56] provides a continuous historical record (1881–1986) of SST anomalies averaged over an irregular region of the equatorial Pacific extending for 5° N–5° S and 150°–90° W. The dataset is obtained from Geographic Data Sharing Infrastructure, College of Urban and Environmental Science, Peking University (http://geodata.pku.edu.cn, (accessed on 8 April 2023)).

### 4.2. Experiment Setup

Generally, a three-layer MLP network structure is selected for experiments. The activation function in the hidden layer is given by the sigmoid function, while the activation function in the output layer is a linear function. Then, we focus on discussing the optimal neuron size in the hidden layer. The Akaike information criterion based on the generalized degrees of freedom (AIC_g_) is employed to evaluate the performance of the MLP networks. During the model training process, the SGD optimization algorithm is used; the momentum is set to 0.9, the weight decay is 0.0005, and the initial learning rate is 0.01.

In the following section, we first describe the results of phase space reconstruction and then give the results of the network model optimization. The performance of the models obtained is also analyzed in detail.

Two measurement methods, the maximum Lyapunov exponent (MLE) and the mean square error (MSE) of the time series, were considered in the experiment for evaluating the experimental results.

Maximum Lyapunov Exponents (MLE):

The Lyapunov exponent can quantitatively characterize the chaotic attractor by measuring the sensitivity of the chaotic orbit to the original condition. In general, a time sequence is considered chaotic only when the *MaxLE* is positive based on the phase track. Li and Xu proposed an estimation method of *MaxLE* by the local Jacobian matrix of the nonlinear system [57]. In the reconstructed delay phase space, the local Jacobian matrix at this point is rewritten in the form
$$A_{n}=\left[\begin{array}{ccccc} 0 & 1 & 0 & \cdots & 0\\ 0 & 0 & 1 & \cdots & 0\\ \vdots & \vdots & \vdots & \ddots & \vdots \\ 0 & 0 & 0 & \cdots & 1\\ b_{1,n} & b_{2,n} & b_{3,n} & \cdots & b_{d,n} \end{array}\right]$$
where
(43)$$b_{k,n}=\frac{\partial f\left(X_{n}\right)}{\partial x_{n,k}}$$
In this case, the nonlinear function is the three-layer MLP network, and Equation (43) can be rewritten as
(44)$$b_{k,n}=\sum_{m=1}^{M}w_{m}^{\left(2\right)}o_{m,n}\left(1-o_{m,n}\right)w_{m,k}^{\left(1\right)}$$

Mean square error (MSE)

The mean square error (MSE) is defined by
(45)$$MSE=\frac{1}{N}\sum_{j=1}^{N}{\left(y_{n}-{\hat{y}}_{n}\right)}^{2}$$
where *y_n_* and ${\hat{y}}_{n}$ are the MLP’s output and expected output of the *n*th input, and *N* is the number of samples.

### 4.3. Phase Space Reconstruction

For phase space reconstruction, the two parameters of the delay interval *τ*^*^ and embedding dimension *m*^*^ should be determined. The delay interval *τ*^*^ is determined by the mutual information method, and the embedded dimension *m*^*^ is selected by the Cao method. Figure 2 shows the variation curve of mutual information with delay interval *τ* on four datasets. The first minimum point or inflection point of the four curves is selected as the final delay interval. In the Henon map dataset, when *τ* = 11, the curve enters the stationary region, so the time interval is selected as 11. In the Lorenz dataset, the curve reaches the first minimum point when *τ* = 13, so the time interval of the Lorenz dataset is selected as 13. In the sunspot dataset and the equatorial Pacific sea-surface temperature dataset, the first minimum is reached at *τ* = 35 and *τ* = 24, respectively, so the time interval is chosen as 35 and 24, respectively. Figure 3 shows the change of the two indicators *E*_1_ and *E*_2_ of the CAO method with the embedding dimension on four datasets, and we select the definite state where *E*_1_ reaches a stationary state and *E*_2_ does not. As shown in the figure, in the four datasets of the Henon map, Lorenz, sunspots, and equatorial Pacific sea-surface temperature, the embedding dimensions are selected as 5, 3, 7, and 4, respectively.

### 4.4. Analysis for AIC_g_

We trained different networks on the Lorentz dataset by randomly initializing the parameters (here, the same network topology is used but with different network weights). The number of nodes in the hidden layer of the network is set to 20, and 10 networks are obtained by training from 10 random initialized parameters. Table 1 shows the comparison results of generalized degrees of freedom (GDF), Akaike information criterion (AICg), and maximum Lyapunov exponents (MLE) for these 10 networks on the Lorenz dataset. The results show that the models with small AICg values prefer small MLE values. The scatterplots for MLE and AICg also show the same trend (Figure 4). In other words, AICg is a good criterion for selecting a good network for chaotic time series.

### 4.5. Model Optimization

In order to find the optimal network topology for chaotic time series, we select 20 candidate network structures, that is, the number of neurons in the hidden layer from 11 to 30. Each network is trained by randomly selecting initial weights from 10 groups, and the model parameter with the smallest AIC_g_ was selected. Figure 5 shows the changes of AIC_g_ with hidden layer sizes on the Henon map, Lorenz equation, sunspot, and SST datasets, respectively. For example, as can be seen in Figure 5a, the smallest AIC_g_ value is obtained when the number of hidden layer nodes is 19, which is the selected optimal structure. Similarly, for the Lorenz equation, sunspots, and equatorial Pacific sea-surface temperature datasets, the smallest AIC_g_ values were obtained with the number of hidden nodes being 27, 19, and 17.

The maximal Lyapunov exponents (MLEs) are calculated by the above method for the four chaotic time series. Three MLP networks—one built by the optimal neuron size and the other two networks built by 20 and 25 neurons, respectively–are employed to estimate the MLE for the four datasets. The two compared networks are optimized by simply minimizing the loss function on MSE. The numerical results are listed in Figure 6. The numerical results show that the network model optimized by our proposed method is significantly better than the two network structures with a fixed number of hidden layer nodes of 20 and 25 on the Lorenz and sunspot datasets, slightly better than the other two types of networks on the Henon map dataset, roughly equivalent to the other two networks on the SST dataset, slightly better than the 25 hidden layer node network, and slightly worse than the 20 hidden layer node network results.

In order to further compare the performance of our proposed method, we investigate the multi-step prediction effect of the above three types of networks on the four datasets, namely, the optimized network by our proposed method, and two MLP networks with 20 and 25 neurons in the hidden layer, respectively. They are trained by the same dataset. The comparison results are shown in Figure 7. In Figure 7a, we show that the optimized MLP network provides significantly better prediction results than the other two MLP networks on the Henon map dataset; the optimized MLP network performs the best on all time steps. It also indicates that the AIC_g_ performs well in choosing a good MLP network for a chaotic time series. Figure 7a shows the results of the Lorenz dataset. Except for the fifth time step, the optimized MLP network performs the best among the three networks. The comparison results on two real-world time series are shown in Figure 7c,d. For the first five steps, the MLP networks can predict the time series quite well, but after five steps, the predicting errors become larger. The MLP network selected by the AIC_g_ also behaves much better than the other two MLP networks in nearly all cases. It also demonstrates the benefit of using the AIC_g_ to choose a good MLP network for a chaotic time series.

## 5. Conclusions

Focusing on chaotic time series modeling, this paper proposes an MLP network-based framework. First, the generalized degree of freedom approximation method of MLP networks is derived, and then the Akaike information criterion (AIC_g_) can be calculated. Next, a multilayer perceptron network optimization method for chaotic time series analysis is designed according to AIC_g_. Finally, this paper proposes an overall framework for chaotic time series analysis. To verify the effectiveness of the proposed method, it is applied to four chaotic time series datasets, including two artificial datasets and two actual datasets. Experimental results show that the proposed method can effectively optimize the MLP network, and the selected model has obvious performance advantages. At the same time, the results show that the model selected in this paper can obtain good prediction performance on all four datasets. In future work, this approach could be extended to deep neural network models.

## Figures and Tables

**Figure 1 entropy-25-00973-f001:**
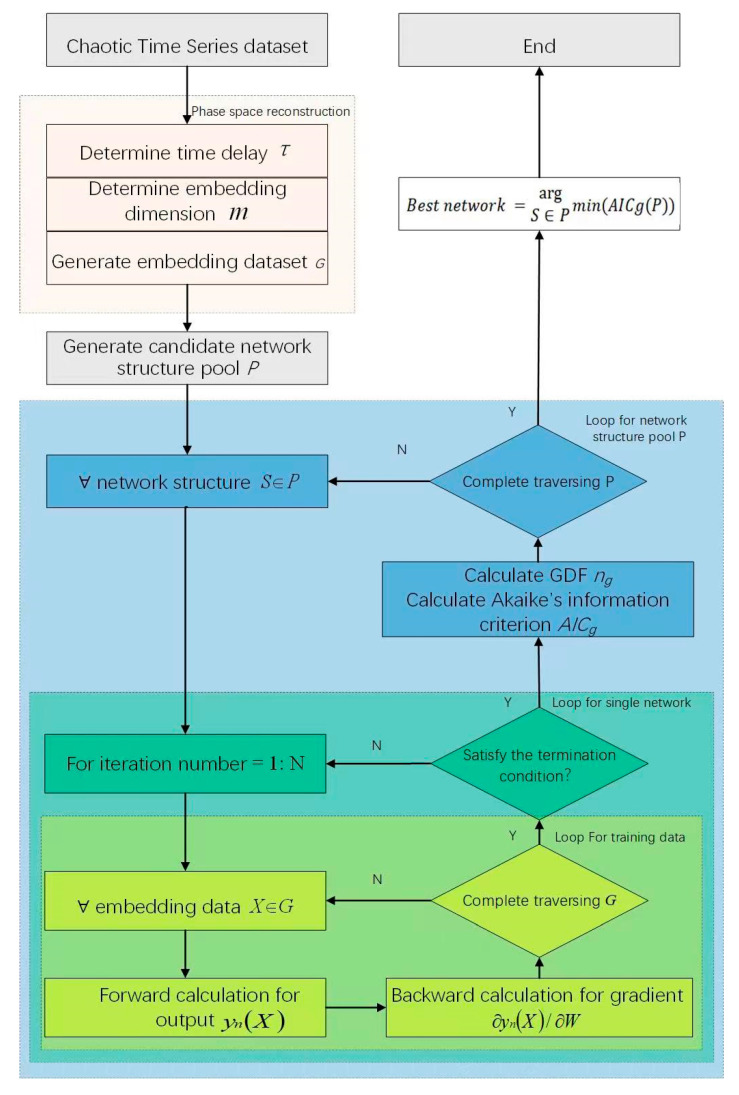
Pipeline of the modeling selection.

**Figure 2 entropy-25-00973-f002:**
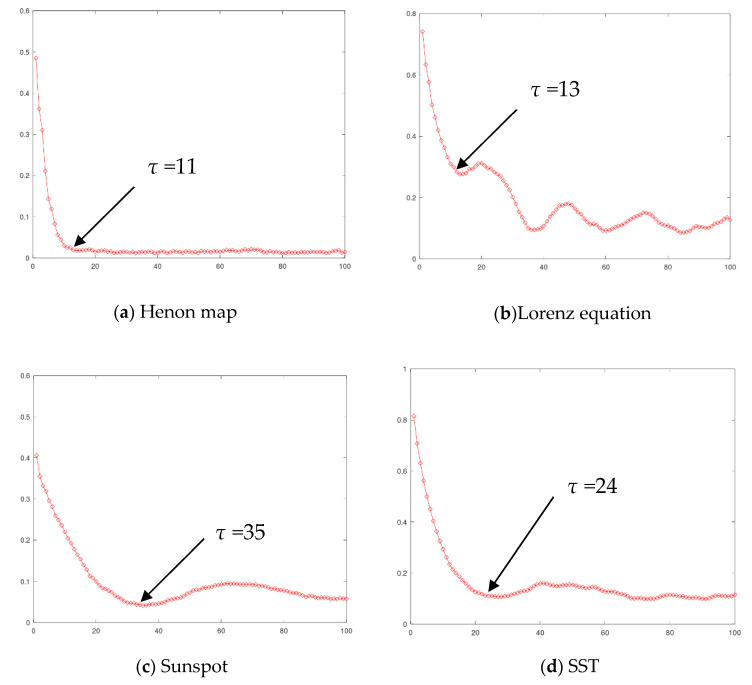
Variation curve of mutual information with delay interval *τ* on datasets of the (**a**) Henon map, the (**b**) Lorenz equation, (**c**) sunspot, and (**d**) SST.

**Figure 3 entropy-25-00973-f003:**
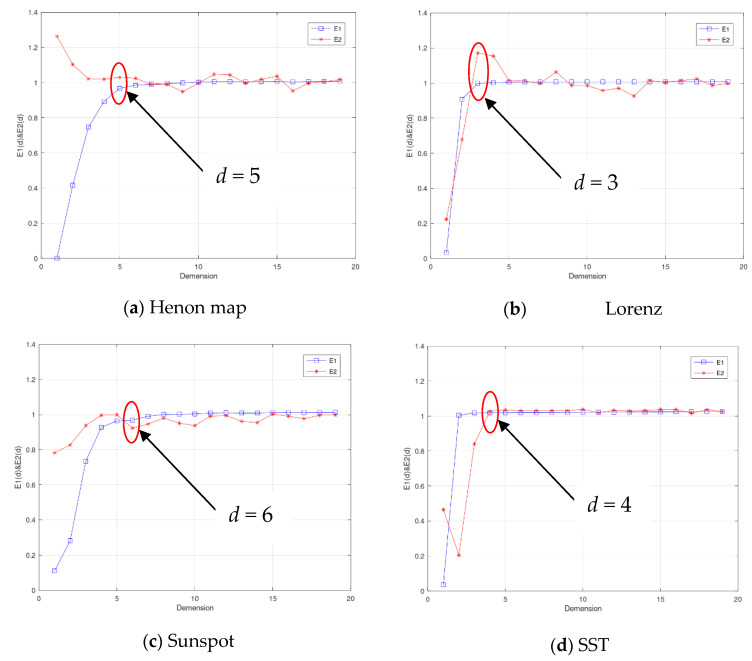
The *E*_1_ and *E*_2_ indicators change curves with the embedded dimension *d* on datasets of the (**a**) Henon map, (**b**) Lorenz equation, (**c**) sunspot, and (**d**) SST.

**Figure 4 entropy-25-00973-f004:**
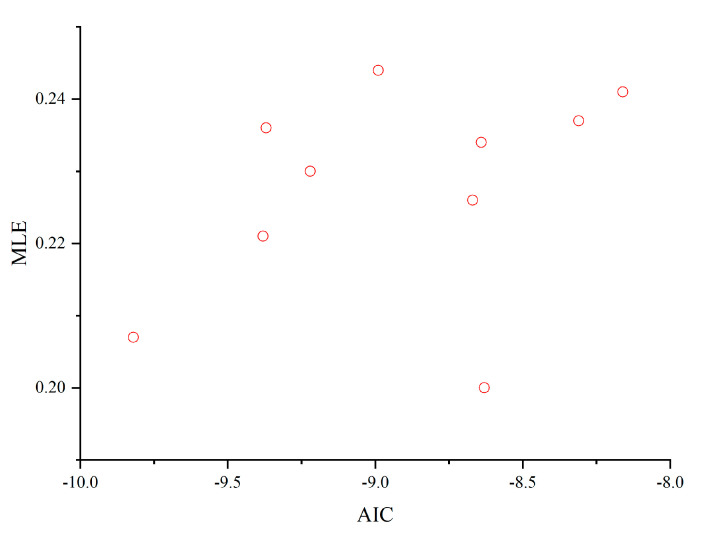
Scatterplot of AICg and MLE on 10 different networks.

**Figure 5 entropy-25-00973-f005:**
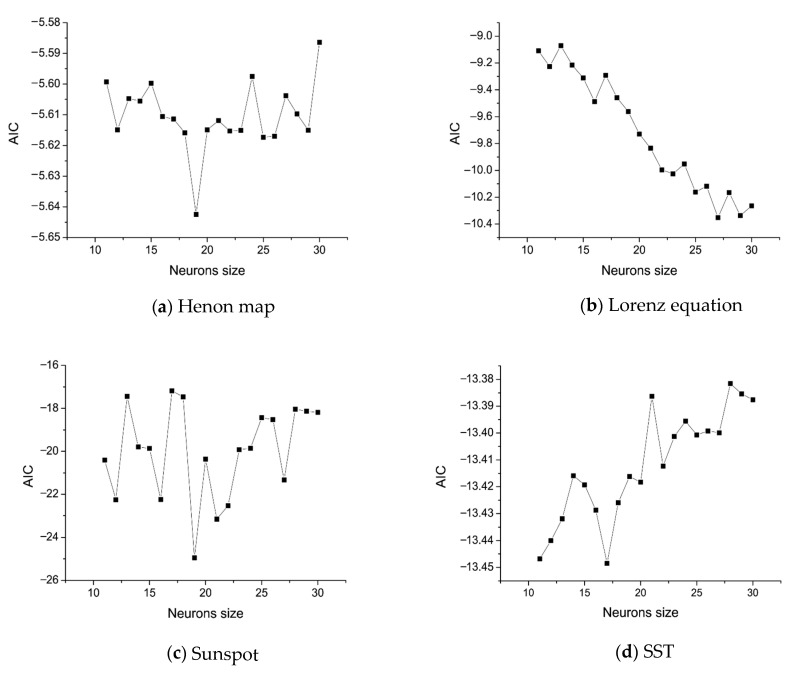
AIC_g_ curves of hidden layer scales on the datasets of the (**a**) Henon map, (**b**) Lorenz equation, (**c**) sunspot, and (**d**) SST.

**Figure 6 entropy-25-00973-f006:**
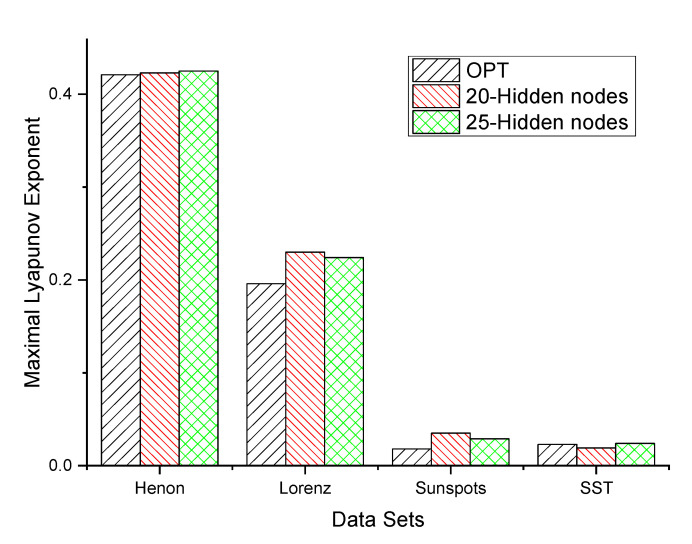
Lyapunov exponent estimation using the MLP model.

**Figure 7 entropy-25-00973-f007:**
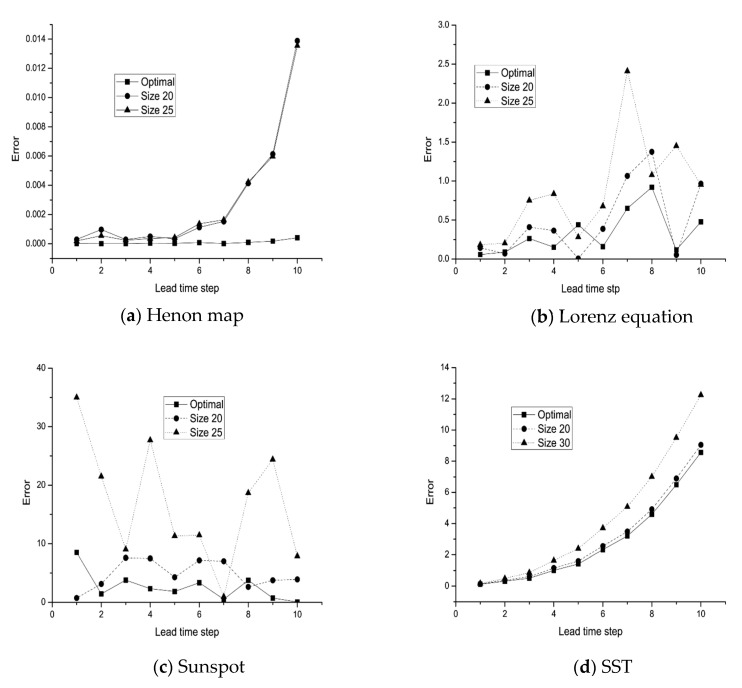
MSE error comparisons on datasets of the (**a**) Henon map, (**b**) Lorenz equation, (**c**) sunspot, and (**d**) SST.

**Table 1 entropy-25-00973-t001:** Comparison results of AIC_g_, GDF, and MLE on 10 networks.

	GDF	AIC	MLE
1	76.00	−9.22	0.230
2	76.00	−9.37	0.236
3	69.93	−8.31	0.237
4	74.01	−8.67	0.226
5	76.00	−8.99	0.244
6	73.60	−9.38	0.221
7	75.98	−8.64	0.234
8	76.00	−9.82	0.207
9	76.00	−8.63	0.200
10	76.00	−8.16	0.241

## Data Availability

The open-access data that support the findings of this study are made publicly available at psl.noaa.gov/gcos_wgsp/Timeseries/SUNSPOT/ (accessed on 5 June 2023) and http://geodata.pku.edu.cn (accessed on 8 April 2023).

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
