# Peer review of "Multilayer Perceptron Network Optimization for Chaotic Time Series Modeling"

_entropy, 2023, doi:10.3390/e25070973_

Round 1

Reviewer 1 Report

Please find the comments in the uploaded file.

Minor editing of English language required.

Reviewer 2 Report

Dear authors!

The paper presents a very interesting work on optimization of the multilayer perceptron network for chaotic system identification.

The work proposes a novel pipeline for model optimization, including Akaike’s information criterion, and three embedded loops of model selection: for training data, for single network and for network structure pool. It is shown that on four test problems, including the Henon map, the Lorenz Equation, the Sunspot and SST datasets, the proposed approach gives fruitful results.

Nevertheless, several remarks should be made.

1) The benefits of using your approach should be quantified. Please give a comparison with a standard optimization routine which includes simply minimizing the loss function on MSE between the result of the network output and training data, or any other optimization routine for MLP in similar problem reported in literature.

2) Please present plots or tables illustrating improving the loss function value during optimization process to show how three-stage pipeline works.

3) Some references are missing. First, please add a brief note in Introduction on other approaches to system identification including symbolic regression, NARMAX models and so on, and why these approaches are inferior to the MLP approach. I recommend the following works to be cited: DOI: 10.1007/s11071-022-07854-0, DOI: 10.1109/WCICA.2006.1712650, other appropriate citations are appreciated. 

Also, there is a number of approaches on reconstructing the phase space. What is the novelty of your particular approach? Please compare it with some previously reported ones, for example, described in a paper DOI:  10.1103/PhysRevE.84.016223

English language is fine, but some stylistic inaccuracies present

Round 2

Reviewer 1 Report

All the comments have been addressed. I think it can be accepted.